# Psychological Resilience and Health-Related Quality of Life in 418 Swedish Women with Primary Breast Cancer: Results from a Prospective Longitudinal Study

**DOI:** 10.3390/cancers13092233

**Published:** 2021-05-06

**Authors:** Åsa Mohlin, Pär-Ola Bendahl, Cecilia Hegardt, Corinna Richter, Ingalill Rahm Hallberg, Lisa Rydén

**Affiliations:** 1Department of Clinical Sciences Lund, Division of Medical History, Lund University, BMC, 221 84 Lund, Sweden; 2Healthcare Center Laröd, Travvägen 27, 252 86 Helsingborg, Sweden; 3Department of Clinical Sciences Lund, Division of Oncology, Lund University, Medicon Village, 223 81 Lund, Sweden; par-ola.bendahl@med.lu.se (P.-O.B.); cecilia.hegardt@med.lu.se (C.H.); 4CREATE Health—Translational Cancer Center, Department of Immunotechnology, Lund University, Medicon Village, 223 81 Lund, Sweden; corinna.richter@immun.lth.se; 5Department of Health Sciences, Lund University, P.O. Box 157, 221 00 Lund, Sweden; ingalill.rahm_hallberg@med.lu.se; 6Department of Clinical Sciences Lund, Division of Surgery, Lund University, Medicon Village, 223 81 Lund, Sweden; lisa.ryden@med.lu.se; 7Department of Surgery, Skåne University Hospital, Södra Förstadsgatan 1, 214 28 Malmö, Sweden

**Keywords:** breast cancer, psychological resilience, health-related quality of life, Connor–Davidson Resilience Scale 25 (CD-RISC25), Short Form Health Survey (SF-36)

## Abstract

**Simple Summary:**

Psychological resilience is an important psychological mechanism that enables a person to successfully handle significant adversities, e.g., a cancer diagnosis. Despite improved prognosis, breast cancer is associated with emotional distress across the trajectory of the disease. This study aimed to investigate psychological resilience and health-related quality of life in Swedish women with breast cancer at diagnosis and one year later. The resilience score declined in the cohort and was associated with health-related quality of life at both time points. Assessment of psychological resilience in breast cancer care might enable the identification of patients in need of intensified rehabilitation to improve their health-related quality of life.

**Abstract:**

Psychological resilience is considered a major protective psychological mechanism that enables a person to successfully handle significant adversities, e.g., a cancer diagnosis. Higher levels of resilience have been associated with higher levels of health-related quality of life (HRQoL) in breast cancer (BC) patients, but research examining the longitudinal process of resilience is limited. The aim of this population-based longitudinal study was to investigate resilience and HRQoL from diagnosis to one year later in 418 Swedish women with primary BC. Resilience was measured with the Connor–Davidson Resilience Scale 25, and HRQoL was measured with the Short Form Health Survey. The participants responded to questions regarding demographic and study-specific variables. Clinicopathological variables were collected from the Swedish National Quality Register for Breast Cancer. The mean score for resilience was 70.6 (standard deviation, SD = 13.0) at diagnosis and 68.9 (SD = 14.0) one year later, *p* < 0.001. The level of trust in the treatment and financial situation demonstrated the greatest association with the change in resilience levels. No oncological treatment modality was associated with a change in resilience levels. HRQoL decreased over time in the cohort. Resilience was positively associated with HRQoL at one year post diagnosis, which demonstrates that resilience is an important factor in maintaining HRQoL.

## 1. Introduction

Breast cancer (BC) is the most common type of cancer in women worldwide [1]. Despite improved prognosis, BC is associated with emotional distress across the trajectory of the disease. This high level of distress may prevent a patient from coping effectively with the disease and its treatments [2]. Psychological resilience (henceforth referred to as resilience), defined as the ability to cope successfully with external stress, is considered to be one of the major protective psychological mechanism that enables a patient to handle this distress [3,4,5]. Resilience is presumed to be an evolving process that allows a patient to adapt and thrive when facing significant adversities [3,6,7]. Previous studies have found higher levels of resilience to be associated with higher levels of health-related quality of life (HRQoL) in women with BC [4,7,8,9]. In this context, resilience reflects the psychological resources that the women can mobilize in order to maintain their HRQoL throughout the BC trajectory. High-resilience patients will more effectively manage their new life situations [3]. Due to its favorable survival rate, HRQoL has become an important outcome measure for BC patients [10,11], and by adding resilience, it might be possible to predict HRQoL and to identify those most in need of additional rehabilitation interventions [5].

Although several studies have reported on resilience among BC patients, most of these studies are cross-sectional with a wide range of time points for the resilience assessments, making it challenging to interpret the results [4,6,8,9,12,13,14,15,16,17,18,19]. However, the resilience levels were often negatively affected by the BC diagnosis, underlining the importance to further study this psychological mechanism to be able to enhance HRQoL in BC patients.

Research examining the dynamics of resilience by collecting longitudinal data from the diagnostic and treatment phases onward is very limited. Since different phases of the cancer trajectory can be demanding for BC patients, longitudinal assessments may identify individual characteristics, illness-related, and treatment-related factors that are positively associated with resilience and thus HRQoL. Social support was previously identified as a main variable positively associated with resilience [4,9,12,15]. If we knew more about how resilience can change over time in BC patients, we could design more individualized short- and long-term interventions for women presenting lower levels of resilience. Due to the dynamic nature of resilience, continuous and longitudinal assessments of resilience are of great importance to improve the knowledge about this protective psychological mechanism among BC patients. Interventions that can contribute to develop resilience might improve the life situations for many women, as higher levels of resilience are associated with better HRQoL.

The present study focuses on changes in resilience and HRQoL from BC diagnosis to one year post diagnosis in a Swedish cohort of 418 women. To the best of our knowledge, this is one of the largest population-based longitudinal studies concerning this topic. This study also extends the resilience research in Sweden, since relatively little is known about resilience in Swedish women with BC.

The Connor–Davidson Resilience Scale 25 (CD-RISC25), the most commonly used resilience scale and one of the resilience scales with the best psychometric properties [20,21,22], was used in this study to assess resilience at the time of the diagnosis and at one year post diagnosis. The Swedish version of the Short Form Health Survey (SF-36) was used to measure HRQoL [23,24].

The present study aimed to investigate the levels of resilience and HRQoL at the time of BC diagnosis and at one year post diagnosis and to investigate the change in resilience levels in relation to the demographic, clinicopathological, and study-specific characteristics in Swedish women with primary BC.

## 2. Materials and Methods

### 2.1. Study Design and Study Cohort

This longitudinal study was performed within the large prospective BC study SCAN-B Resilience (ClinicalTrials.gov Identifier: NCT03430492) [25], which is a part of the Sweden Cancerome Analysis Network—Breast (SCAN-B) initiative [26]. The SCAN-B study is a population-based study that includes almost 90% of all patients newly diagnosed with BC from hospitals in southern Sweden (ClinicalTrials.gov Identifier: NCT02306096) [26,27]. Based on a comparison with women reported in the Swedish National Quality Register for Breast Cancer (NKBC) during the same time period, the SCAN-B cohort provides a good representation of patients diagnosed in the catchment area [28].

Women with primary BC were included between February 2016 and December 2019 in SCAN-B Resilience at Blekinge County Hospital (Karlskrona), Central Hospital Växjö, Hallands Hospital Halmstad, and Helsingborgs Hospital. Karlskrona, Växjö, Halmstad, and Helsingborg are all urban cities. Karlskrona, Växjö, and Halmstad feature more rural areas than Helsingborg, while Helsingborg is a larger, more multi-cultural city. The participants were included at the time they were informed of their BC diagnosis. The inclusion criteria for SCAN-B Resilience were previously described in the published study protocol [25]. Potentially eligible participants were identified by their BC nurses. Patients were first enrolled in SCAN-B. Once enrolled in the main study, they were invited to be enrolled in the substudy, SCAN-B Resilience. The inclusion rate of SCAN-B Resilience was approximately 70%. Participants were given oral and written information about the study before starting any procedures, and informed consent was signed by all participants.

In September 2018, 517 women with newly diagnosed BC were selected for a cross-sectional baseline study [29]. The selected participants presented complete assessments (CD-RISC25 and SF-36) and NKBC data (clinicopathological variables) at diagnosis. The participants completed the assessments, electronically or on paper, right after the cancer consultation in which they were informed about the diagnosis and treatment plan. The diagnostic work-up, including mammography, ultrasound, and biopsy, was performed two to three weeks before the visit to the breast units at the hospitals.

In total, 418 participants presented complete assessments and NKBC data at the follow-up one year post diagnosis and were selected for this longitudinal study (Figure 1). At follow-up, the participants completed the assessments at home, electronically, or on paper. The non-response rate of the present study was 19% (*n* = 99) (Figure 1). Comparisons of the characteristics of the participants of the baseline study (*n* = 517), the participants of this follow-up study (*n* = 418), and the excluded participants (*n* = 99) are presented in Appendix A.

SCAN-B and SCAN-B Resilience were approved by the Regional Ethical Review Board in Lund (Dnr 2009/658, 2010/383, 2012/58, 2013/459, 2015/277, 2015/522, 2016/944, 2017/875, 2017/88) and the Swedish Ethical Review Authority (Dnr 2019-00700, 2019-01351). The approval included access to the NKBC data and administration of the instruments.

### 2.2. Measures

Age and clinicopathological variables, including menstrual status, mode of detection, stage of BC, type of cancer, tumor size, histological type, estrogen receptor (ER) status, progesterone receptor (PR) status, human epidermal growth factor receptor 2 (HER2) status, Ki67 percentage, surgery type, neoadjuvant therapy, and adjuvant therapy, were collected from the NKBC register. Information on almost 100% of Swedish women diagnosed with BC since 2008 is available in the NKBC register [28].

To generate demographic and study-specific variables, the participants answered questions added to the standardized instruments regarding their weight and height (used to calculate their body mass index (BMI)), physical activity, smoking habits, social network, educational level, and financial situation. These variables were chosen due to their known relationships with health outcomes [30]. The question about the ability to pay an unexpected bill of SEK 11,000/EUR 1100 is widely used in Swedish population studies [31]. At follow-up, the participants also responded to study-specific questions regarding their trust in the treatment they received for BC, their satisfaction with the implementation of the treatment they received, and their satisfaction with the staff–patient encounters throughout the treatment process.

The Swedish version of the CD-RISC25 was used to measure resilience. Permission to use the instrument was obtained from the developer. The instrument consists of 25 items (e.g., “Able to adapt to change”, “Think of yourself as a strong person”, and “Tend to bounce back after illness or hardship”) coded on a Likert scale ranging from 0 (“Not true at all”) to 4 (“True nearly all the time”) [20,21]. The total score over the 25 items can thus range from 0 to 100, with higher scores reflecting higher levels of resilience. In previous studies, where resilience was measured in relation to health problems, including various cancer diseases, the CD-RISC25 demonstrated good validity and reliability [21,22]. Recently, a Swedish population-based study was performed, where the mean resilience score was 68.7 among women in a non-clinical population (*n* = 1283, Cronbach’s alpha = 0.92) [7]. However, this non-clinical population was selected based on examining different health aspects focusing on heart and lung diseases. Swedish norm data for CD-RISC25 are not yet available. For the present longitudinal study, the Cronbach’s alpha of the CD-RISC25 was 0.93 in the follow-up cohort.

HRQoL was assessed using the Swedish version of the SF-36, which consists of 36 items (e.g., “Compared to one year ago, how would you rate your general health now?”) [23,24]. The original coding algorithm was used for the items (raw scores transformed into a 0–100 range), where higher scores represented a better HRQoL. The items were grouped into eight domains: physical functioning (PF), role limitations due to physical problems (RP), bodily pain (BP), general health (GH), vitality (VT), social functioning (SF), role limitations due to emotional problems (RE), and mental health (MH). The Swedish version of the SF-36 has been shown to have good validity and reliability, and Swedish norm data are available [23,24]. Permission to use this instrument was obtained from Optum (Optum Circle, Eden Prairie, MN, USA). For this study, the Cronbach’s alpha of the SF-36 was 0.89 in the follow-up cohort.

### 2.3. Statistical Analysis

Descriptive statistics, including means and standard deviations (SDs), were calculated for the demographic, clinicopathological, and study-specific variables, as well as for the total CD-RISC25 and SF-36 scores. For age, the median was also calculated. A change (Δ) in the CD-RISC25 score was defined as the score at one year post diagnosis minus the score at diagnosis.

To investigate the differences between the included and excluded participants, the demographic and clinicopathological characteristics and CD-RISC25 mean score were compared between the included (*n* = 418) and excluded participants (*n* = 99). An independent samples *t*-test was used for these comparisons. No systematic differences between these groups were found, and all unadjusted *p*-values were >0.05.

An independent samples *t*-test and a one-way ANOVA were used to compare the mean score and mean ΔCD-RISC25 across the groups of demographic, clinicopathological, and study-specific variables. Model diagnostics were assessed and revealed no deviations that questioned the use of these tests.

A paired-samples *t*-test was used to compare the mean scores for the CD-RISC25 and SF-36 at baseline and follow-up. An independent samples *t*-test was used to compare the mean score for SF-36 in the study cohort with the Swedish norm data.

Multiple linear regression analyses with stepwise backward selection, *p* > 0.157 for removal (a threshold equivalent to the Akaike information criterion (AIC) for model selection [32]), was applied to identify a set of variables with independent predictive value after adjusting for the baseline CD-RISC25 score. For the categorical variables, dummy variables were created, with the following coded as Group 1: being postmenopausal, having screening-detected BC, having stage 0 and I BC, having received no adjuvant chemotherapy, having received adjuvant endocrine therapy, having received adjuvant radiotherapy, engaging in physical activity every day and at least three to four times a week, living with a child/children, having post-secondary education and a PhD, being able to pay an unexpected bill of SEK 11,000/EUR 1100, having greater trust in the treatment, having greater satisfaction with the implementation of the treatment, and having greater satisfaction with the staff–patient encounters throughout the treatment process.

Uni- and multivariable linear regression analyses were also used to investigate the associations between CD-RISC25 and SF-36 at follow-up. Potential confounders were examined in these analyses by adjusting for the demographic, clinicopathological, and study-specific variables in the multivariable regression analyses.

Significance for all statistical tests was set at 0.05 or less, but because no adjustment for multiple testing was performed, the level of evidence for a specific test against the null hypothesis should be interpreted with some caution. The statistical programs used were IBM SPSS Statistics (version 25.0, IBM Inc., Armonk, NY, USA) and Stata version 16 (StataCorp, College Station, TX, USA).

## 3. Results

### 3.1. Study Cohort

In total, 418 women with primary BC were included in this longitudinal study (Figure 1). The participants ranged in age from 31 to 85 years with a median age of 64 years (Table 1). The study population had similar clinicopathological characteristics to the Swedish BC population diagnosed during the same time period according to the NKBC register [28].

The results of the study specific questionnaire regarding BMI, smoking habits, physical activity, social network, educational level, and financial situation are presented in Table 2. Importantly, 58% reported a very high level of trust in the treatment they are receiving or received for their BC, 66% reported a very high level of satisfaction with the implementation of the treatment they receive/received, and 76% reported a very high level of satisfaction with the staff–patient encounters throughout the treatment process.

Additional participant characteristics are described in Appendix A.

### 3.2. Psychological Resilience

The mean resilience score was 70.6 (SD = 13.0) at diagnosis and 68.9 (SD = 14.0) at one year post diagnosis (*p* < 0.001) (Table 3). The distribution of the change (Δ) in the resilience score between the two time points is depicted in Figure 2.

Postmenopausal women had lower levels of resilience at one year post diagnosis than premenopausal women (Table 1). Those with a greater level of trust in the treatment, a greater level of satisfaction with the implementation of the treatment, and a greater level of satisfaction with the staff–patient encounters throughout the treatment process had higher levels of resilience at one year post diagnosis (Table 2). Participants who were more physically active also tended to have higher resilience. Additionally, those who were able to pay an unexpected bill of SEK 11,000/EUR 1100 tended to have higher resilience (Table 2).

Screening-detected BC was associated with less of a negative change in the resilience score compared to a symptomatic diagnosis (Table 1). Women who were able to pay an unexpected bill of SEK 11,000/EUR 1100 also had less of a negative change in their resilience score, as did those with a greater level of trust in the treatment (Table 2).

Analysis of the other demographic, clinicopathological, and study-specific variables revealed no significant relationship with resilience (Table 1 and Table 2). Notably, no oncological treatment modality was associated with the changes in resilience score in our cohort.

### 3.3. Health-Related Quality of Life

As presented in Table 3, the women in this study had lower mean scores for several domains of HRQoL (i.e., GH, SF, RE, and MH) at diagnosis compared to the Swedish norm data. The mean score of BP was higher in the study cohort, indicating lower levels of pain in the cohort compared to the norm data.

The cohort, furthermore, had lower mean scores for even more domains of HRQoL at one year post diagnosis (Table 3). Except for BP, the physical aspects of SF-36—PF, RP, and GH—were lower in the cohort compared to the norm data. At one year post diagnosis, all psychological aspects of SF-36—VT, SF, RE, and MH—were lower in the cohort.

### 3.4. Predictors of the Change in Resilience Score

Multiple linear regression analyses with backward selection were conducted to identify predictors of the change in the resilience score between the two time points (Table 4). The stepwise selection started with a full model, including the centered baseline resilience score and the following list of potential dichotomous or dichotomized variables: menstrual status, mode of detection, stage of BC, adjuvant therapies, physical activity, social network, education level, financial situation, trust in the treatment, satisfaction with the treatment, and satisfaction with the staff–patient encounters. In addition to the baseline resilience score, in backward elimination (*p* > 0.157 for removal), the following variables were selected: a greater level of trust in the treatment (*β* = 6.415, *p* = 0.002), and being able to pay an unexpected bill of SEK 11,000/EUR 1100 (*β* = 3.649, *p* = 0.034). Thus, when comparing two groups of patients with the same baseline resilience score and the same financial situation but with different levels of trust in the treatment, the group with a greater level of trust will on average have a 6.415 unit smaller drop in the resilience score between the two time points compared to those with a lower level of trust. Those who are able to pay an unexpected bill of SEK 11,000/EUR 1100 will on average have a 3.649 unit smaller drop in the resilience score compared to those that are not able to pay a bill of that amount, adjusted for the baseline resilience score and the level of trust in the treatment. The final model explains 12% of the variance in the change in the resilience score.

### 3.5. Uni- and Multivariable Regression Analyses between Psychological Resilience and Health-Related Quality of Life at One Year Post Diagnosis

Uni- and multivariable linear regression analyses were conducted to test the associations between resilience and each of the eight domains of HRQoL at one year post diagnosis. Potential confounding variables were investigated by adjusting for the demographic, clinicopathological, and study-specific variables (i.e., the variables described above assumed to be associated with resilience) in the multivariable regression model (Table 5).

The uni- and multivariable regression analyses demonstrated strong evidence for associations between resilience and all domains of HRQoL. Higher levels of resilience were associated with higher levels of HRQoL at one year post diagnosis. The relative changes in the *β*-coefficients for each of the HRQoL domains were minor, indicating that the adjustment factors were not strong confounders for the association between resilience and HRQoL at one year post diagnosis.

## 4. Discussion

The present study investigated psychological resilience and HRQoL in a large cohort of Swedish women with primary BC collected at the time of diagnosis to one year follow-up in relation to the demographic, clinicopathological, and study-specific characteristics. To the best of our knowledge, this is the first population-based longitudinal study conducted among Swedish BC patients and one of the largest studies to date on longitudinal resilience and HRQoL assessments in BC patients. The results of the present study extend previous research by demonstrating a strong link between longitudinal resilience measurements and HRQoL. Resilience and HRQoL decreased during the first year after diagnosis in the Swedish BC cohort. The scores were lower than those of the general population norm data at both time points. Importantly, no oncological treatment modality was associated with changes in resilience levels.

This study included a cohort of 418 Swedish women with a mean resilience score of 70.6 (SD = 13.0) at diagnosis and 68.9 (SD = 14.0) at one year post diagnosis. In line with prior BC studies, where the mean resilience score varied between 54.7 and 74.7 [4,6,9,12,15,17,18,33], the study cohort herein presented lower levels of resilience compared to the normative data reported by Connor and Davidson [20]. The inconsistency in resilience levels between the BC studies might be due to differences in cohort characteristics and, possibly more important, the timing of the CD-RISC25 assessment across the BC trajectory, since resilience is an evolving process [3,4]. Patients with higher levels of resilience are considered to cope more effectively with the disease and associated treatments and to recover their health better than patients with lower levels of resilience [4]. However, one year seems insufficient for recovery, as the mean resilience score was lower at one year post diagnosis in the cohort.

In this longitudinal study, the findings of a mean decrease in resilience score of 1.7 units from diagnosis to one year later might be an indicator for unmet rehabilitation needs among these Swedish BC patients. Since low resilience levels have been linked to worse health outcomes [34], the early assessment of resilience would be of putative importance to practitioners and healthcare providers. Enhanced rehabilitation, including psychosocial support for less resilient patients, might be a way to improve the health-related outcomes among this group of women.

In the study cohort, postmenopausal women had lower resilience levels both at diagnosis and at one year later compared to premenopausal women [29]. This could indicate that older BC patients have lower resilience levels in a Swedish context; however, the change in resilience score was not related to menstrual status. Moreover, prior research regarding age and resilience in different study settings is inconsistent. As discussed by Wu et al., both positive and negative relationships between age and resilience have been demonstrated in previous studies [12] alongside the lack of a significant relationship, as reported by Huang et al. and Padilla-Ruiz et al. [4,19]. The association between age and resilience among Swedish BC patients needs to be explored in further research.

The change in resilience score was also related to each patient’s financial situation, namely, the ability to pay an unexpected bill of SEK 11,000/EUR 1100. The inability to work can be one of the many stressors associated with BC and its treatment [10,35]. Economic concerns in this study were related to a greater drop in the mean resilience score. Wu et al. found that a higher monthly family income was associated with higher resilience levels in their BC cohort [12], further supporting our findings.

Women with symptomatically-detected BC also had a more negative change in their resilience scores compared to women with screening-detected BC in our Swedish cohort. As demonstrated in our baseline study [29], women with screening-detected BC tended to have lower resilience levels at diagnosis. It is reasonable to assume that these asymptomatic women were initially more shocked by their BC diagnosis compared to the women who had already felt a symptomatic lump in their breasts when they contacted a healthcare provider. Therefore, the resilience levels among the women with screening-detected BC could have been more strongly affected at baseline; moreover, the difference between the groups evened out over the year until the follow-up assessment. It may also be possible that the group of women with symptomatically-detected BC included those with more advanced BC, affecting the patients’ resilience levels more negatively. Research regarding how resilience is affected depending on how BC is detected remains limited. Additional research on this subject could potentially confirm whether some women with screening-detected BC initially need more psychosocial support. Huang et al. reported that the resilience levels decreased with an increase in the tumor stage of the disease and the courses of adjuvant therapy [4]. However, we found no relationship in our study between resilience and tumor stage or between resilience and different neoadjuvant/adjuvant therapy modalities, which is consistent with the results of Padilla-Ruiz et al. and Wu et al. [12,19].

In this study, we found strong evidence for higher resilience levels in women with a greater level of trust in the treatment, a greater level of satisfaction with the implementation of the treatment, and a greater level of satisfaction with the staff–patient encounters. One cannot reliably assess the direction of the association between these variables: It is possible that highly resilient patients are more prone to feeling greater levels of trust and satisfaction. Of the variables included in this study, the level of trust in the treatment demonstrated the greatest association with a change in resilience score. With an increasing level of trust, the change in the resilience score became less negative during the year after diagnosis. It is tempting to assume that a low level of trust and/or satisfaction reflects low-resilience patients who received inadequate rehabilitation during the BC treatment, resulting in even lower resilience levels at one year post diagnosis. Again, the assessment of resilience might be useful in a clinical setting for the early identification of women in need of additional rehabilitation interventions. Improved resilience, captured as a higher resilience score, can help a patient to cope successfully with adversities across the cancer trajectory [4].

The Swedish BC cohort had lower mean scores in all domains of HRQoL evaluated by the SF-36 at one year post diagnosis compared with the Swedish norm data [24]. Except for the BP domain, there was strong evidence for lower mean scores in all seven other domains—PF, RP, GH, VT, SF, RE, and MH—at one year post diagnosis. Already at diagnosis, several domains of HRQoL (i.e., GH, SF, RE, and MH) were lower in the cohort compared to the Swedish normative data. The fear of BC during the diagnostic work-up likely already had an impact on the HRQoL levels at the time of consultation for the BC diagnosis, reflected by the psychological aspects of HRQoL being more affected at that time point. Intense BC treatments can induce many side effects over the cancer trajectory, which could explain the decline that also occurred in the physical aspects of HRQoL at one year post diagnosis. Lower levels of HRQoL in BC patients compared to the population norms have been demonstrated in previous studies [11,36], but studies exploring resilience in relation to HRQoL are limited. Despite differences in the HRQoL instruments, our findings are in agreement with the results reported by Ristevska-Dimitrovska et al., Zhang et al., and Tu et al., demonstrating that the levels of resilience are positively related to the levels of HRQoL in BC patients [8,9,16]. The positive associations between resilience and the eight domains of HRQoL remained significant after adjustment for the demographic, clinicopathological, and study-specific characteristics. These findings underscore the importance of identifying low-resilience patients, as they may experience decreasing HRQoL across their demanding cancer journeys. Highly resilient patients may be better at protecting their HRQoL, since they are likely better able to cope with the negative distress caused by BC [4,16].

The limitations of the present study include the non-response rate of 19%. Some of the excluded participants may have had lower/higher scores of resilience and/or HRQoL than the included participants. Another limitation is the lack of information on prediagnosis psychological and/or physical disorders among the participants. Interpretation of our results was, complicated by the lack of Swedish norm data for the CD-RISC25, as well as norm data for the CD-RISC25 in a BC setting. Moreover, it was also challenging to compare our results to those in the literature due to the variety of contexts and the timing of the measurements in prior resilience studies. More frequent and coherent assessments of resilience across the cancer trajectory may increase our understanding of the resilience process in BC patients. Qualitative research may also enrich our understanding of resilience.

The strengths of the present study include our large population-based BC sample with longitudinal data from diagnosis to one year post diagnosis regarding resilience and HRQoL. Importantly, all of the participants completed the instruments at the same time points. Both early and coherent assessments of resilience have not been previously described, enabling us to decipher the changes in resilience levels between well-defined time points among BC patients.

## 5. Conclusions

The results of this prospective population-based longitudinal study provide evidence that resilience is an important factor in maintaining HRQoL among women with BC. This study is one of the largest to date on longitudinal resilience and HRQoL among BC patients and the first conducted in a Swedish BC setting. Resilience decreased over the year after diagnosis in the Swedish BC cohort. The level of trust in the treatment and financial situation demonstrated the greatest association with the change in resilience levels implicating that psychosocial support are of importance. No oncological treatment modality was associated with a change in resilience levels. HRQoL also decreased over time in the cohort. The scores of resilience and HRQoL were lower than those of the general population norm data at both time points. These results indicate that the participants did not fully recover over the first year after diagnosis and may indicate unmet rehabilitation needs among these patients. Resilience was positively associated with HRQoL. Our findings highlight the importance for the early identification of patients with low resilience, as these patients may experience an even greater decrease in HRQoL across the demanding cancer trajectory compared to patients with high resilience. If further research can establish clinically relevant thresholds for CD-RISC25, the assessment of resilience might provide a way to identify BC patients in need of additional rehabilitation interventions. Psychological interventions should aim to enhance resilience in BC patients, since our findings clearly demonstrate a strong link between longitudinal resilience and HRQoL in these patients.

## Figures and Tables

**Figure 1 cancers-13-02233-f001:**
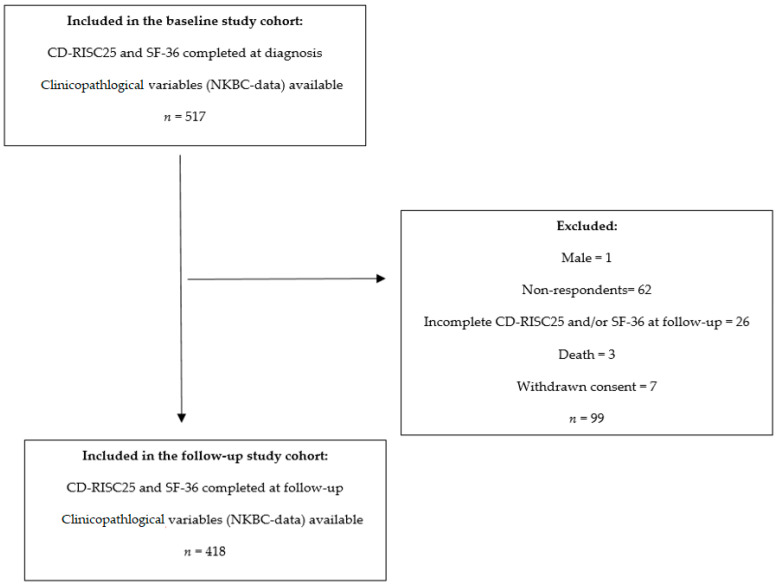
Flow chart of the study cohort. Abbreviations: CD-RISC25, Connor-Davidson Resilience Scale 25; SF-36, Short Form Health Survey; NKBC, Swedish National Quality Register for Breast Cancer.

**Figure 2 cancers-13-02233-f002:**
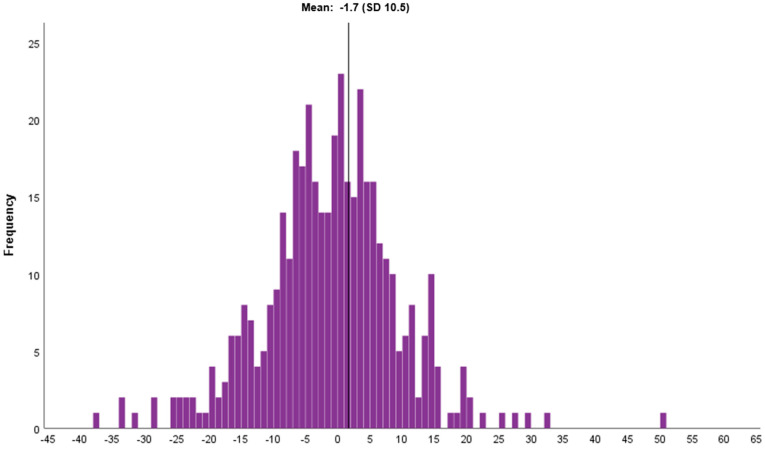
Histogram of the change in psychological resilience score (CD-RISC25) (score at 1 year post diagnosis minus score at diagnosis) (*n* = 418). Abbreviations: CD-RISC25, Connor-Davidson Resilience Scale 25; SD, standard deviation.

**Table 1 cancers-13-02233-t001:** Mean score for psychological resilience (CD-RISC25) at 1 year post diagnosis, and mean change in psychological resilience score between 1 year post diagnosis and diagnosis (delta-CD-RISC25) according to demographic and clinicopathological characteristics.

Variables		*n*	%	CD-RISC25 at 1 Year Post Diagnosis		delta-CD-RISC25 (Score at 1 Year Post Diagnosis Minus Score at Diagnosis)	
				Mean (SD)	*p*-Value ^A^	Mean (SD)	*p*-Value ^A^
Follow-up cohort	-	418		68.9 (14.0)	*-*	−1.7 (10.5)	*-*
Study site	Halmstad	135	32	68.5 (13.7)	0.587	−1.6 (10.6)	0.961
	Helsingborg	20	5	71.7 (19.2)		−2.4 (9.0)	
	Karlskrona	120	29	69.8 (13.6)		−1.3 (10.6)	
	Växjö	143	34	68.1 (13.8)		−1.9 (10.6)	
Age (years)	Mean (SD): 62 (11)	-	-	-	-	-	-
	≤64 (median)	214	51	69.9 (12.9)	0.134	−1.7 (10.0)	0.954
	>64 (median)	204	49	67.8 (15.0)		−1.6 (11.0)	
Menstrual status	Premenopausal	74	19	72.3 (13.7)	0.024	−2.1 (11.2)	0.699
	Postmenopausal	325	81	68.2 (14.1)		−1.5 (10.4)	
	Unknown	19					
Mode of detection	Screening	269	65	69.0 (13.5)	0.764	−0.8 (10.2)	0.022
	Symptomatic	148	35	68.6 (14.9)		−3.3 (10.9)	
	Unknown	1					
Stage	0	24	6	69.9 (13.8)	0.114	0.8 (7.0)	0.691
	I	272	65	68.0 (14.3)		−1.9 (10.7)	
	II	117	28	70.4 (13.0)		−1.7 (10.7)	
	III	3	1	83.7 (18.3)		0.3 (2.1)	
	Unknown	2					
Type of breast surgery	Breast-conserving	304	73	68.3 (14.0)	0.136	−1.7 (10.7)	0.949
	Mastectomy	114	27	70.5 (13.9)		−1.6 (10.0)	
	Unknown	0					
Immediate breast reconstruction	Yes	25	7	67.6 (12.7)	0.540	−1.9 (7.9)	0.776
	No	333	93	69.3 (13.9)		−1.1 (10.3)	
	Unknown	60	-	-		-	
Type of axillary surgery	Sentinel node	323	79	68.3 (14.0)	0.200	−1.9 (10.9)	0.690
	Axillary dissection	35	9	69.5 (13.6)		−1.4 (10.9)	
	Sentinel node + axillary dissection	52	13	72.0 (13.9)		−0.76 (7.7)	
	Unknown	8	-	-			
Neoadjuvant chemotherapy	Yes	20	5	69.7 (12.0)	0.801	−3.2 (11.2)	0.505
	No	398	95	68.8 (14.1)		−1.6 (10.5)	
	Unknown	0					
Adjuvant chemotherapy	Yes	151	36	69.6 (14.0)	0.506	−2.2 (9.7)	0.307
	No	267	64	68.6 (14.1)		−1.1 (10.5)	
	Unknown	0					
Adjuvant endocrine therapy	Yes	271	65	69.1 (14.0)	0.693	−1.2 (10.6)	0.186
	No	147	35	68.5 (14.2)		−2.6 (10.3)	
	Unknown	0					
Adjuvant bisphosphonate therapy	Yes	57	14	71.6 (11.7)	0.074	−2.1 (8.4)	0.736
	No	361	86	68.5 (14.8)		−1.6 (10.8)	
	Unknown	0					
Adjuvant antibody therapy ^B^	Yes	45	11	67.6 (12.4)	0.517	−4.2 (11.1)	0.093
	No	373	89	69.0 (14.2)		−1.4 (10.4)	
	Unknown	0					
Adjuvant radiotherapy	Yes	334	80	68.9 (13.7)	0.946	−1.6 (10.5)	0.764
	No	84	20	68.8 (15.0)		−2.0 (10.4)	
	Unknown	0	-	-		-	

Notes: ^A^ Independent-samples *t*-test for comparison of means in two groups, one-way analysis of variance for comparison of three or more group means. ^B^ HER2-targeted therapy. Abbreviations: CD-RISC25, Connor-Davidson Resilience Scale 25; SD, standard deviation.

**Table 2 cancers-13-02233-t002:** Mean score for psychological resilience (CD-RISC25) at 1 year post diagnosis and mean change in psychological resilience score between 1 year post diagnosis and diagnosis (delta-CD-RISC25) according to demographic and study-specific characteristics.

Variables		*n*	%	CD-RISC25 at 1 Year Post Diagnosis		delta-CD-RISC25 (Score at 1 Year Post Diagnosis Minus Score at Diagnosis)	
				Mean (SD)	*p*-Value ^A^	Mean (SD)	*p*-Value ^A^
Follow-up cohort	-	418	-	68.9 (14.0)	-	−1.7 (10.5)	-
BMI	Underweight (<18.5)	1	0.2	80.0	0.605	−2.0	0.920
	Normal-weight (18.5–24.9)	173	42	68.4 (13.8)		−1.5 (9.0)	
	Overweight (25.0–29.9)	152	37	69.8 (14.4)		−2.2 (12.2)	
	Obese (≥30)	84	20	68.1 (14.0)		−1.2 (10.5)	
	Unknown	8					
Cigarette smoker	I smoke every day	32	8	69.0 (11.1)	0.378	−1.1 (10.5)	0.705
	I smoke sometime during the month	6	1	78.2 (8.9)		−6.0 (10.3)	
	I have previously smoked every day but do not smoke on a regular basis today	134	32	69.3 (14.5)		−1.3 (11.8)	
	I have never smoke regularly	246	59	68.4 (14.1)		−1.9 (9.8)	
Physical activity (at least 30 min)	Every day	121	29	71.0 (13.4)	0.052	−0.6 (10.1)	0.259
	At least 3–4 times a week	145	35	69.0 (14.3)		−3.0 (9.4)	
	At least 1–2 times a week	102	24	68.3 (13.6)		−1.0 (11.4)	
	Less than once a week	50	12	64.6 (14.8)		−1.9 (12.5)	
Social network	Living alone	85	20	68.2 (15.9)	0.198	−2.5 (10.2)	0.092
	Living with child/children < 18 years old only	5	1	76.0 (21.1)		−6.2 (7.0)	
	Living with adult/adults and child/children < 18 years old	53	13	72.1 (13.6)		−4.2 (9.4)	
	Living with adult/adults only	275	66	68.3 (13.2)		−0.8 (10.8)	
Educational level	Primary school < 9 years	55	13	68.2 (16.6)	0.213	−0.9 (12.9)	0.856
	Primary school completed	52	12	64.9 (15.0)		−2.4 (10.6)	
	Upper secondary education	84	20	68.8 (13.0)		−1.1 (11.2)	
	Post-secondary education <2 years	39	9	68.9 (14.6)		−2.8 (10.7)	
	Post-secondary education ≥ 2 years	182	44	70.4 (12.8)		−1.6 (9.3)	
	PhD (doctoral education)	6	1	65.3 (20.4)		−5.3 (12.4)	
Financial situation	Able to pay an unexpected bill of SEK 11,000/EUR 1100	378	90	69.4 (13.6)	0.050	−1.3 (10.3)	0.022
	Unable to pay an unexpected bill of SEK 11,000/EUR 1100	40	10	63.9 (16.7)		−5.3 (11.9)	
Trust in the treatment	Not at all	5	1	69.8 (18.9)	<0.001	−0.4 (5.5)	0.047
	To a lesser extent	1	0.2	61.0		−17.0	
	To some extent	19	5	57.5 (14.0)		−7.5 (9.8)	
	To a large extent	151	36	66.1 (14.4)		−2.1 (11.5)	
	To a very high extent	242	58	71.6 (12.9)		−0.9 (9.9)	
Satisfaction with the implementation of the treatment	Not at all	2	0.5	58.0 (14.1)	<0.001	−3.5 (3.5)	0.174
	To a lesser extent	2	0.5	72.0 (15.6)		−5.0 (17.0)	
	To some extent	13	3	56.5 (16.4)		−6.1 (9.2)	
	To a large extent	126	30	65.4 (14.9)		−3.0 (10.6)	
	To a very high extent	275	66	71.1 (12.8)		−0.8 (10.5)	
Satisfaction with the staff–patient encounters	Not at all	0	0	-	0.005	-	0.181
	To a lesser extent	4	1	69.5 (13.5)		−8.0 (13.6)	
	To some extent	15	4	59.8 (18.6)		−3.1 (12.6)	
	To a large extent	83	20	65.8 (14.6)		−3.4 (9.1)	
	To a very high extent	316	76	70.1 (13.4)		−1.1 (10.7)	

Notes: ^A^ Independent-samples *t*-test for comparison of means in two groups, one-way analysis of variance for comparison of three or more group means. Abbreviations: CD-RISC25, Connor-Davidson Resilience Scale 25; BMI, body mass index; SD, standard deviation.

**Table 3 cancers-13-02233-t003:** Mean scores for health-related quality of life (SF-36) at diagnosis (baseline) and at 1 year post diagnosis (follow-up) and norm values of SF-36. Mean scores for psychological resilience (CD-RISC25) at diagnosis and at 1 year post diagnosis.

Variables	Baseline (BL) Mean (SD)	Follow-Up (FU)Mean (SD)	Difference between BL Data and FU Data *p*-Values ^A^	Swedish Norm Data Mean (SD)	Difference between BL Data and Norm Data *p*-Values ^B^	Difference between FU Data and Norm Data *p*-Values ^B^
**SF-36 ^C^**						
Physical functioning	85.9 (17.6)	80.5 (20.3)	<0.001	86.2 (20.4)	0.771	<0.001
Role—physical	83.6 (33.3)	69.6 (40.8)	<0.001	81.6 (33.1)	0.237	<0.001
Bodily pain	82.3 (20.0)	70.3 (22.6)	<0.001	72.7 (26.5)	<0.001	0.073
General health	71.4 (19.1)	68.6 (20.9)	0.001	75.1 (22.7)	0.001	<0.001
Vitality	68.2 (23.0)	62.0 (24.3)	<0.001	66.7 (23.2)	0.205	<0.001
Social functioning	85.1 (21.7)	82.9 (23.5)	0.062	87.5 (20.8)	0.024	<0.001
Role—emotional	77.5 (36.1)	77.2 (37.6)	0.904	84.0 (30.9)	<0.001	<0.001
Mental health	70.5 (21.5)	76.7 (18.4)	<0.001	79.6 (19.4)	<0.001	0.003
**CD-RISC25 ^D^**	70.6 (13.0)	68.9 (14.0)	0.001	Not available	-	-

Notes: ^A^ Paired-samples *t*-test. ^B^ Independent-samples *t*-test. ^C^ Range 0–100, high values represent high functioning. ^C,D^ Range 0–100, high values represent high levels of resilience. Abbreviations: CD-RISC25, Connor-Davidson Resilience Scale 25; SF-36, Short Form Health Survey; BL, baseline; FU, follow-up; SD, standard deviation.

**Table 4 cancers-13-02233-t004:** Multiple linear regression analyses with stepwise backward elimination ^A^. Predictors of the change in psychological resilience score (score at 1 year post diagnosis minus score at diagnosis) after adjustment for baseline resilience score.

Final Model	β	95% CI	*p*-Values
Constant	−10.908	−15.872 to −5.934	-
Centered baseline CD-RISC25 score	−0.254	−0.329 to −0.178	<0.001
Greater level of trust in the treatment	6.415	2.292 to 10.538	0.002
Able to pay an unexpected bill of SEK 11 000/EUR 1100	3.649	0.278 to 7.019	0.034

Variables entered: Centered baseline resilience score, menstrual status, mode of detection, stage of breast cancer, adjuvant chemotherapy, adjuvant endocrine therapy, adjuvant radiotherapy, physical activity, social network, education level, financial situation, trust in the treatment, satisfaction with the implementation of the treatment, and satisfaction with the staff–patient encounters. Notes: ^A^
*p*-value > 0.157 for removal. R2, goodness-of-fit for the final model = 0.122. Abbreviations: CD-RISC25, Connor-Davidson Resilience Scale 25; β, Beta coefficient; 95% CI, 95% confidence interval.

**Table 5 cancers-13-02233-t005:** Uni- and multivariable linear regression analyses between psychological resilience (CD-RISC25) and health-related quality of life (SF-36) at 1 year post diagnosis.

	Unadjusted Models				Adjusted Models ^A^		
Variables	CD-RISC25 at 1 Year Post Diagnosis				CD-RISC25 at 1 Year Post Diagnosis		
SF-36	β	95% CI	*p*	R^2^	β	95% CI	*p*
Physical functioning	0.339	0.203 to 0.475	<0.001	0.054	0.250	0.067 to 0.433	0.008
Role—physical	0.767	0.496 to 1.039	<0.001	0.069	0.764	0.384 to 1.144	<0.001
Bodily pain	0.283	0.130 to 0.437	<0.001	0.031	0.370	0.152 to 0.587	0.001
General health	0.649	0.519 to 0.778	<0.001	0.189	0.605	0.423 to 0.787	<0.001
Vitality	0.722	0.569 to 0.874	<0.001	0.172	0.655	0.446 to 0.864	<0.001
Social functioning	0.503	0.349 to 0.658	<0.001	0.090	0.374	0.157 to 0.592	0.001
Role—emotional	0.812	0.565 to 1.059	<0.001	0.091	0.753	0.406 to 1.100	<0.001
Mental health	0.626	0.514 to 0.737	<0.001	0.226	0.577	0.420 to 0.735	<0.001

Notes: ^A^ Adjusted for baseline resilience score, menstrual status, mode of detection, stage of breast cancer, adjuvant chemotherapy, adjuvant endocrine therapy, adjuvant radiotherapy, physical activity, social network, education level, financial situation, trust in the treatment, satisfaction with the implementation of the treatment, and satisfaction with the staff–patient encounters. Abbreviations: CD-RISC25, Connor-Davidson Resilience Scale 25; SF-36, Short Form Health Survey; β, Beta coefficient; 95% CI, 95% confidence interval; *p*, *p*-value; R^2^, goodness-of-fit.

## Data Availability

The anonymized datasets used and/or analyzed during the current study are available from the corresponding author upon reasonable request.

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
