# Peer review of "Psychological Resilience and Health-Related Quality of Life in 418 Swedish Women with Primary Breast Cancer: Results from a Prospective Longitudinal Study"

_cancers, 2021, doi:10.3390/cancers13092233_

Round 1

Reviewer 1 Report

  1. Mohlin and hes colleagues have successfully investigated psychological resilience and health-related quality of life in women with breast cancer at diagnosis and one year later. In general, this is an important contribution on the topic because  the assessment of psychological resilience in breast cancer care might enable the identification of patients in need of intensified rehabilitation to improve their health-related quality of life.  Hence, this study would provide evidence based information on the identification of patients to improve  their health-related quality of life. However, I have minor suggestions:
  2. Introduction: I suggest that the authors should justify and clearly state the rationale and significance of the development of this study. It would be better to revisit the background section and indicate the study development’s significance and how this study could be applied to the formation of prevention initiatives.
  3. The statement of the problem in the introduction is not enough with regards to the Psychological Resilience. I recommend citing published studies.
  4. Methods: the methods clearly presented as the study is ongoing and the methodology is sufficiently mentioned, including study design and stages of the research.
  5. The conclusions should be slightly expanded, they are too short and do not seem to outline concrete results (which, instead, have been obtained).

Author Response

Mohlin and hes colleagues have successfully investigated psychological resilience and health-related quality of life in women with breast cancer at diagnosis and one year later. In general, this is an important contribution on the topic because the assessment of psychological resilience in breast cancer care might enable the identification of patients in need of intensified rehabilitation to improve their health-related quality of life.  Hence, this study would provide evidence based information on the identification of patients to improve their health-related quality of life.

Answer:

We thank the reviewer for these comments.

However, I have minor suggestions:

Introduction: I suggest that the authors should justify and clearly state the rationale and significance of the development of this study. It would be better to revisit the background section and indicate the study development’s significance and how this study could be applied to the formation of prevention initiatives.

Answer:

Thank you for pointing this out, parts of the Introduction section have been rewritten according to this comment. We have clarified the importance of studying the relationship between psychological resilience and health-related quality of life (HRQoL) among breast cancer (BC) patients and elaborated on the unique longitudinal study design on page 2, line 53 - 89

Resilience is presumed to be an evolving process that allows a patient to adapt and thrive when facing significant adversities [3, 6, 7]. Previous studies have found higher levels of resilience to be associated with higher levels of health-related quality of life (HRQoL) in women with BC [4, 7-9]. In this context, resilience reflects the psychological resources that the women can mobilize in order maintain their HRQoL throughout the BC trajectory. High resilient patients will more effectively manage their new life situations [3]. Due to its favorable survival rate, HRQoL has become an important outcome measure for BC patients [10, 11], and by adding resilience, it might be possible to predict HRQoL and to identify those most in need of additional rehabilitation interventions [5].

Although several studies have reported on resilience among BC patients, most of these studies are cross-sectional with a wide range of time points for the resilience assessments, making it challenging to interpret the results [4, 6, 8, 9, 12-19]. However, the resilience levels were often negatively affected by the BC diagnosis, underlining the importance to further study this psychological mechanism to be able to enhance HRQoL in BC patients.

Research examining the dynamics of resilience by collecting longitudinal data from the diagnostic and treatment phases onward is very limited. Since different phases of the cancer trajectory can be demanding for BC patients, longitudinal assessments may identify individual characteristics, illness-related and treatment-related factors that are positively associated with resilience and thus HRQoL. Social support was previously identified as a main variable positively associated with resilience [4, 9, 12, 15]. If we knew more about how resilience can change over time in BC patients, we could design more individualized short- and long-term interventions for women presenting lower levels of resilience. Due to the dynamic nature of resilience, continuous and longitudinal assessments of resilience are of great importance to improve the knowledge about this protective psychological mechanism among BC patients. Interventions that can contribute to develop resilience might improve the life situations for many women, as higher levels of resilience are associated with better HRQoL.

The present study focuses on changes in resilience and HRQoL from BC diagnosis to one year post diagnosis in a Swedish cohort of 418 women. To the best of our knowledge, this is one of the largest population-based longitudinal studies concerning this topic. This study also extend the resilience research in Sweden, since relatively little is known about resilience in Swedish women with BC.

The statement of the problem in the introduction is not enough with regards to the Psychological Resilience. I recommend citing published studies.

Answer:

Thank you for this comment, we have rewritten parts of the Introduction section according to this comment and the comment under query 2 – please also refer to the reply above. We have clarified the important role resilience plays in the BC patients’ adjustment to their illness on p 2, line 63-67.

 Although several studies have reported on resilience among BC patients, most of these studies are cross-sectional with a wide range of time points for the resilience assessments, making it challenging to interpret the results [4, 6, 8, 9, 12-19]. However, the resilience levels were often negatively affected by the BC diagnosis, underlining the importance to further study this psychological mechanism to be able to enhance HRQoL in BC patients.

Moreover, we have described previous cross-sectional studies within this research field, and highlight the need of longitudinal studies since resilience is assumed to be a dynamic process on p 2, line 79-83.

Due to the dynamic nature of resilience, continuous and longitudinal assessments of resilience are of great importance to improve the knowledge about this protective psychological mechanism among BC patients. These interventions that can contribute to develop resilience might improve the life situations for many women, as higher levels of resilience are associated with better HRQoL.

Methods: the methods clearly presented as the study is ongoing and the methodology is sufficiently mentioned, including study design and stages of the research.

Answer:

Thank you for this comment.

The conclusions should be slightly expanded, they are too short and do not seem to outline concrete results (which, instead, have been obtained).

Answer:

Thank you for highlighting the need for clarification, we have added the following in the conclusions section on p 14-15, lines 469-484:

The results of this prospective population-based longitudinal study provide evidence that resilience is an important factor in maintaining HRQoL among women with BC. This study is one of the largest to date on longitudinal resilience and HRQoL among BC patients and also the first conducted in a Swedish BC setting. Resilience decreased over the year after diagnosis in the Swedish BC cohort. The level of trust in the treatment and financial situation demonstrated the greatest association with the change in resilience levels implicating that psychosocial support are of importance. No oncological treatment modality was associated with a change in resilience levels. HRQoL also decreased over time in the cohort. The scores of resilience and HRQoL were lower than those of the general population norm data at both time points. These results indicate that the participants did not fully recover over the first year after diagnosis and may indicate unmet rehabilitation needs among these patients. Resilience was positively associated with HRQoL. Our findings highlight the importance for the early identification of patients with low resilience, as these patients may experience an even greater decrease in HRQoL across the demanding cancer trajectory compared to patients with high resilience. If further research can establish clinically relevant thresholds for CD-RISC25, the assessment of resilience might provide a way to identify BC patients in need of additional rehabilitation interventions. Psychological interventions should aim to enhance resilience in BC patients, since our findings clearly demonstrate a strong link between longitudinal resilience and HRQoL in these patients.

Reviewer 2 Report

The authors present a contribution about resilience in BC patients. The study is straightforward and confirmatory in nature, verifying findings from previous study. A few suggestions:

  • The introduction is very brief. The paper would benefit from a more in depth presentation of the contribution Resilience has on the health of BC patients. Also, the authors could try to highlight more how their research question contribute to the state of the art.
  • lines 118-125: I would ask the authors to evaluate leaving out these lines, as they are repetition of information presented elsewhere (i.e. fig.1)
  • line 194-202: Please rephrase this section, the list of variables is hard to read and follow. 
  • line 219-229: I suggest the authors provide the information in this section in a table and/or refer directly to the supplementary material. 
  • The authors assert in the statistical analysis section that model diagnostics were checked. In the tables though, it emerges that the compared groups have wildly different sample size. To me this is alarming, as homogeneous sample size is required by most parametric tests. I suggest the authors specify which diagnostics they run and the results they obtained, or to specify which corrections they used to overcome this issue. Please also add VIF and other diagnostics for the regression model. Please also address sample sizes of the analyzed groups in the limitation section. 
  • The use of Deltas should be better justified by the authors; baseline scores could be used as covariate in ANOVA models to obtain more robust analysis and results. 

Together with reworking the introduction, the author should try to highlight how their findings add something new to the literature in the discussion and conclusion section. 

Author Response

The authors present a contribution about resilience in BC patients. The study is straightforward and confirmatory in nature, verifying findings from previous study.

Answer:

We thank the reviewer for these comments.

 A few suggestions:

The introduction is very brief. The paper would benefit from a more in depth presentation of the contribution Resilience has on the health of BC patients. Also, the authors could try to highlight more how their research question contribute to the state of the art.

Answer:

Thank you for pointing this out. Reviewer 1 made similar comments in his/her report – please see our detailed reply above under query 2 and 3, reviewer 1. Parts of the Introduction section have been rewritten according to these comments on page 2, line 53-89.

Resilience is presumed to be an evolving process that allows a patient to adapt and thrive when facing significant adversities [3, 6, 7]. Previous studies have found higher levels of resilience to be associated with higher levels of health-related quality of life (HRQoL) in women with BC [4, 7-9]. In this context, resilience reflects the psychological resources that the women can mobilize in order maintain their HRQoL throughout the BC trajectory. High resilient patients will more effectively manage their new life situations [3]. Due to its favorable survival rate, HRQoL has become an important outcome measure for BC patients [10, 11], and by adding resilience, it might be possible to predict HRQoL and to identify those most in need of additional rehabilitation interventions [5].

Although several studies have reported on resilience among BC patients, most of these studies are cross-sectional with a wide range of time points for the resilience assessments, making it challenging to interpret the results [4, 6, 8, 9, 12-19]. However, the resilience levels were often negatively affected by the BC diagnosis, underlining the importance to further study this psychological mechanism to be able to enhance HRQoL in BC patients.

Research examining the dynamics of resilience by collecting longitudinal data from the diagnostic and treatment phases onward is very limited. Since different phases of the cancer trajectory can be demanding for BC patients, longitudinal assessments may identify individual characteristics, illness-related and treatment-related factors that are positively associated with resilience and thus HRQoL. Social support was previously identified as a main variable positively associated with resilience [4, 9, 12, 15]. If we knew more about how resilience can change over time in BC patients, we could design more individualized short- and long-term interventions for women presenting lower levels of resilience. Due to the dynamic nature of resilience, continuous and longitudinal assessments of resilience are of great importance to improve the knowledge about this protective psychological mechanism among BC patients. Interventions that can contribute to develop resilience might improve the life situations for many women, as higher levels of resilience are associated with better HRQoL.

The present study focuses on changes in resilience and HRQoL from BC diagnosis to one year post diagnosis in a Swedish cohort of 418 women. To the best of our knowledge, this is one of the largest population-based longitudinal studies concerning this topic. This study also extend the resilience research in Sweden, since relatively little is known about resilience in Swedish women with BC.

lines 118-125: I would ask the authors to evaluate leaving out these lines, as they are repetition of information presented elsewhere (i.e. fig.1)

Answer:

We have deleted this redundant information. These alterations are shown by 'track changes' in the manuscript on p 3, line 133-136.

line 194-202: Please rephrase this section, the list of variables is hard to read and follow. 

Answer:

We consider it to be necessary to describe our dummy variables in order to make it possible for the readers to repeat our analysis.

line 219-229: I suggest the authors provide the information in this section in a table and/or refer directly to the supplementary material. 

Answer:

This section is a summary of our variables presented in Table 1b and is now substantially shortened. These variables cannot be found in the Supplementary Table S1.

The results of the study-specific questionnaire regarding BMI, smoking habits, physical activity, social network, educational level and financial situation are presented in Table 1b.

The authors assert in the statistical analysis section that model diagnostics were checked. In the tables though, it emerges that the compared groups have wildly different sample size. To me this is alarming, as homogeneous sample size is required by most parametric tests. I suggest the authors specify which diagnostics they run and the results they obtained, or to specify which corrections they used to overcome this issue. Please also add VIF and other diagnostics for the regression model. Please also address sample sizes of the analyzed groups in the limitation section. 

Answer:

We don´t agree with the reviewer while  equal, or nearly equal, sample sizes in two groups are  not a requirement for a two-sample t-test to be valid. The same holds true for ANOVA-tests evaluating the evidence for differences in means across three or more independent groups. Maybe, the reviewer meant homogeneity of variances? That might be an issue, but it was not in the present study.

Assumptions for multiple linear regression modelling were evaluated by visual inspection of normal probability plots of the residuals and scatterplots of residuals versus fitted values. These plots showed no problems what so ever for the model presented in Table 3. The diagnostic plots for the 16 models presented in Table 4 show that the model assumptions are well met in both uni- and multivariable analysis for the outcome variables Bodily pain, General Health, Vitality and Mental health. For Physical functioning, Role-physical, Social functioning and Role-emotional, we see a bimodal pattern in the residuals in both uni- and multivariable analysis and for Physical functioning also heteroscedasticity, i.e. decreasing variance of the residuals with increasing predicted value. These violations of model assumptions will not bias the estimated slopes, but the standard errors, which are based on the normal distribution, might be too large or too small. Since all the reported P-values are <0.01 and all but one ≤0.001, we regard the result as robust to the magnitude of model deviations observed. Model diagnostics can be added either in the paper or in an appendix, but in our opinion this will not add substantially to the interpretation of the results.

The VIFs were close to 1.0 for the model presented in Table 3 and 2.08 or lower in the eight multiple linear regression models presented in Table 4, indicating that multicollinearity is no problem in these models.

The use of Deltas should be better justified by the authors; baseline scores could be used as covariate in ANOVA models to obtain more robust analysis and results. 

Answer:

The longitudinal design of the present study is one of its strengths, allowing us to study the change in CD-RISC25 and predictors of this change.

To use baseline score as a covariate in ANOVA models is an excellent idea. We used this strategy for the model presented in Table 3 which is clearly clarified in the legend of variables. The same idea could have been applied to the tests of equal change across categories in the rightmost column in Table 1a, but we decided to stick to P-values from a simple model without adjustment here because the test in the adjusted model does not compare the actual means presented.

Together with reworking the introduction, the author should try to highlight how their findings add something new to the literature in the discussion and conclusion section. 

Answer:

We believe that our study makes an important contribution to the literature because to the best of our knowledge, this study is one of the largest population-based longitudinal studies published to date regarding the association between psychological resilience and HRQoL among women with breast cancer. It is also the first longitudinal study conducted in Sweden regarding this association.

We have clarified this in the Introduction section on p 2, line 86-89:

To the best of our knowledge, this is one of the largest population-based longitudinal studies concerning this topic. This study also extend the resilience research in Sweden, since relatively little is known about resilience in Swedish women with BC.

We have furthermore clarified this in the Discussion section on p 12, line 349-351:

To the best of our knowledge, this is the first population-based longitudinal study conducted among Swedish BC patients and is also one of the largest studies to date on longitudinal resilience and HRQoL assessments in BC patients.

We have rewritten parts of the Conclusion section to better describe our main and important findings –  on p 14-15 line 469-484 – please see our detailed reply above under query 5, reviewer 1.

The results of this prospective population-based longitudinal study provide evidence that resilience is an important factor in maintaining HRQoL among women with BC. This study is one of the largest to date on longitudinal resilience and HRQoL among BC patients and also the first conducted in a Swedish BC setting. Resilience decreased over the year after diagnosis in the Swedish BC cohort. The level of trust in the treatment and financial situation demonstrated the greatest association with the change in resilience levels implicating that psychosocial support are of importance. No oncological treatment modality was associated with a change in resilience levels. HRQoL also decreased over time in the cohort. The scores of resilience and HRQoL were lower than those of the general population norm data at both time points. These results indicate that the participants did not fully recover over the first year after diagnosis and may indicate unmet rehabilitation needs among these patients. Resilience was positively associated with HRQoL. Our findings highlight the importance for the early identification of patients with low resilience, as these patients may experience an even greater decrease in HRQoL across the demanding cancer trajectory compared to patients with high resilience. If further research can establish clinically relevant thresholds for CD-RISC25, the assessment of resilience might provide a way to identify BC patients in need of additional rehabilitation interventions. Psychological interventions should aim to enhance resilience in BC patients, since our findings clearly demonstrate a strong link between longitudinal resilience and HRQoL in these patients.

Reviewer 3 Report

Cancer diagnosis always increases stress for both the patient
and his Families. Psychological
 resilience  is  considered a  major  
protective  mechanism  that enables a person to successfully
handle significant adversities.The aim of the study is formulated
correctly. The subject of the manuscript is interesting
and it is worth conducting such research in patients with
other histological diagnoses of the tumor.
Tables and figures are made carefully and facilitate
the analysis of the results
  .

Author Response

Cancer diagnosis always increases stress for both the patient and her families. Psychological resilience is considered a major protective mechanism that enables a person to successfully handle significant adversities. The aim of the study is formulated correctly. The subject of the manuscript is interesting and it is worth conducting such research in patients with other histological diagnoses of the tumor. Tables and figures are made carefully and facilitate the analysis of the results.

Answer:

We thank the reviewer for these comments.